

# Large−Eddy Simulation of Turbulent Flux Patterns over Oasis Surface

Bangjun Cao[1,2], Yaping Shao[2], Xianyu Yang[1,2], Xin Yin[2], Shaofeng Liu[3]

[1]School of Atmospheric Sciences, Chengdu University of Information Technology, Chengdu, 610225, China
5 [2]Institute for Geophysics and Meteorology, University of Cologne, Cologne, 50923, Germany
[3]Southern Marine Science and Engineering Guangdong Laboratory (Zhuhai), School of Atmospheric Sciences, Sun Yat-sen University, Zhuhai, China.

*Correspondence to*: Xianyu Yang (xyang@cuit.edu.cn)

10 **Abstract.** Large-eddy simulation (LES) is widely used for studying surface fluxes over heterogeneous areas, but the parameterizations based on the Monin-Obukhov Similarity Theory (MOST) often used violate the basic assumptions of the very theory and generate inconsistencies with the LES model closures. Experience shows that LES simulated surface fluxes are strongly dependent on model resolution. Here, we propose a novel scheme for turbulent flux estimates in LES models, which computes the fluxes locally using the LES sub-grid closure scheme (here a $k−l$ scheme) and constrained on the 15 macroscopic scale using the MOST. In comparison with several other schemes, the new scheme performs better for various types of land surfaces tested. The surface fluxes estimated with the new scheme are compared with the field measurements over an oasis surface at different height levels. Other quantities related to surface energy balance, including net radiation, ground heat flux and surface skin temperature, simulated using the new scheme are also found to be consistent with the measurements. Sensitivity tests show that the uncertainty of MOST flux vary little with horizontal resolution, while the 20 macroscopic constraint of MOST increases with horizontal resolution.

## 1 Introduction

Surface fluxes, which characterize the exchanges of energy, mass and momentum between the surface and atmosphere, serve as the lower boundary conditions for atmospheric model simulations. How to estimate the fluxes is a central task of Land Surface Models (LSMs). In almost all existing LSMs, they are parameterized via a network of aerodynamic resistances 25 estimated using the Monin-Obukhov Similarity Theory (MOST) (Monin and Obukhov, 1954; Yang et al., 2008) which assumes the atmospheric boundary layer is stationary and horizontally homogeneous.

Research on the parameterization of land surface processes over heterogeneous surfaces has been in progress since the late 1990s (e.g., Giorgi and Avissar, 1997). It is known that surface heterogeneity has both aggregation and dynamic effects on surface fluxes. The heterogeneity of land surface properties (e.g. albedo) leads to the spatial variability of atmospheric and 30 land surface state variables, and the aggregation effect arises because of the nonlinear dependency of the fluxes on the state



variables. Turbulent eddies generated by surface heterogeneity result in the spatial variation of eddy diffusivity and viscosity and hence that of fluxes, known as the dynamic effect. Depending on the scale of surface heterogeneity, these effects gradually decrease with height to a level known as the blending height (Mason, 1988). The "mosaic" (e.g., Ament and Simmer, 2006) and parameter hierarchy method (e.g., Oleson et al., 2007) are widely used to provide estimates of surface

fluxes over heterogeneous areas, but it remains a major challenge to adequately represent surface heterogeneity in numerical weather prediction (NWP) and climate models.

In recent years, large-eddy simulation (LES) models have been developed to explicitly simulate heterogeneous land surface processes to provide the basis for the parameterization of land surface heterogeneity in large-scale models. In LES models (e.g., Deardorff, 1978; Moeng, 1984; Cao et al., 2018), turbulence is divided into grid-resolved large eddies and sub-grid

eddies in the inertial subrange. The effects of sub-grid eddies are commonly quantified using a sub-grid closure scheme, such as the Smagorinsky closure (Smagorinsky, 1963) or the *k-l* closure (Deardorff, 1980). Early LES models are not coupled with LSM and instead, land surface forcing is pre-specified (e.g., Maronga and Raasch, 2013). In recent years, LES models coupled with LSM have been widely used to study atmospheric turbulence over idealized (e.g., Patton et al., 2005) and natural (e.g., Huang and Margulis, 2010; Shao et al., 2013) heterogeneous surfaces. Several temporal and spatial averaging

methods have been developed to calculate the grid-resolved turbulent fluxes, e.g., phase averaging over repeated one-dimensional (e.g., stripes) or two-dimensional (e.g., checkerboard) surface patches (e.g., Patton et al., 2005), and ensemble averaging method (Maronga et al., 2013).

However, the transfer of knowledge from LES to large-scale models, e.g., NWP (presently spatial resolution ~ 10 km) and climate (~ 100 km) models, is hampered by two major problems related to the scale differences. First, in a large-scale model,

surface fluxes associated with a heterogeneous surface are represented by only one or a few grid cells, while LES uses much finer spatial resolution (~10 m). The surface fluxes represented by one (or a few) grid cells in a large-scale model is required to match in theory those from the LES (Svensson et al., 2010; Zhang et al., 2018; Zhou et al., 2021). Second, how to estimate surface fluxes in LES models is an unsolved problem. To our knowledge, in most LES models, LES sub-grid surface fluxes are computed via a LSM designed for large-scale models, namely, based on the MOST framework. However, this approach

has profound deficiencies: (1) the stationarity and horizontal homogeneity assumptions which underpin the MOST generally do not apply to LES sub-grid processes; (2) the MOST similarity functions are empirically derived based time (~ 15-60 min) and spatial (~ 1-10 km) averages from observations in boundary layers over homogeneous surfaces, which are much larger than the time and length scales of the large eddies; (3) the near-surface diffusivity and viscosity estimated from the MOST are commonly different from those estimated by turbulent closure schemes, causing inconsistencies between the closure

scheme and boundary condition in LES. Comparative studies (e.g. Svensson et al., 2010) on mesoscale and LES models have shown that LES models obviously underestimate sensible heat flux during daytime and overestimate it in night time, if the mesoscale models are used as reference.

Shao et al. (2013) suggested a sub-grid flux calculation scheme for LES by estimating eddy diffusivity and viscosity from the closure scheme, rather than from the MOST, but this approach does not warrant the consistency with the MOST when



the LES estimated fluxes are computed for a scale at which MOST is expected to work well. Here, we propose a novel scheme for surface flux calculation applicable to LES models. This scheme consists of two components. The scheme first computes LES sub-grid fluxes using the eddy viscosity and eddy diffusivity estimated from the LES closure scheme without invoking the MOST, taking into consideration the local turbulence characteristics. It then applies a macroscopic constraint such that the fluxes averaged over the LES domain, which corresponds to a scale for which the MOST applies, satisfies the

MOST. The new scheme thereby remedies the above-mentioned deficiencies and makes the transfer of knowledge from LES to large-scale models plausible.

In the arid and semi-arid regions of Northwest China, numerous oases with various land use types exist. The heterogeneous oasis surfaces strongly affect the spatial and temporal patterns of surface fluxes (Liu et al., 2016) and offer ideal test cases for examining the performances of the new scheme. Here, the "Heihe Watershed Allied Telemetry Experimental Research"

(HiWATER, Li et al., 2013) site at Zhangye (38.78°N, 100.49°E, 1594.00 m Above Sea Level (ASL)) is selected for the development and test of the new scheme. Several other schemes are also tested for comparison.

## 2 New Surface Flux Scheme for LES Model

The new scheme is proposed, referred to as the MOST-r scheme, including two components. First, surface fluxes are estimated locally using (but not limited to) the $k−l$ closure scheme, without invoking the MOST similarity functions. Second,

a macroscopic constraint is applied to the LES simulated fluxes using the MOST (Fig. 1). The Weather Reasearch and Forecast (WRF) LES model is chosen for the development, and the $k−l$ scheme (Deardorff, 1980) is chosen as the LES sub-grid closure scheme.

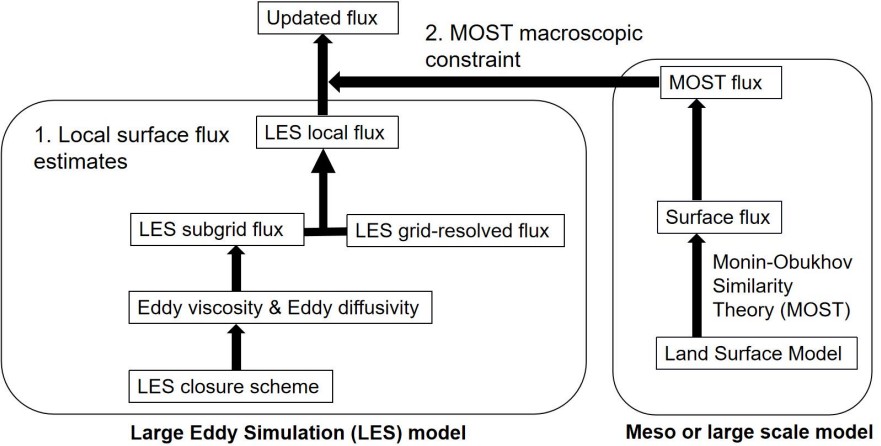

**Figure 1.** Schematic diagram of the MOST-r scheme.



## 2.1 Local surface flux estimates


In LES models, a flux includes the contributions of grid-resolved and sub-grid eddies, e.g., for sensible and latent heat fluxes, $H_{\text{les}}$ and $LE_{\text{les}}$,

$$H_{\text{les}} = H_{\text{les,g}} + H_{\text{les,sg}}, \tag{1}$$

$$LE_{\text{les}} = LE_{\text{les,g}} + LE_{\text{les,sg}}, \tag{2}$$

where $H_{\text{les,g}}$ and $LE_{\text{les,g}}$ are grid-resolved fluxes derived from the simulated vertical velocity $w$, temperature $T$, and specific humidity $q$, *i.e.*,

$$H_{\text{les,g}} = \rho c_{\text{p}} w T, \tag{3}$$

$$LE_{\text{les,g}} = \rho L w q, \tag{4}$$

where $\rho$ is air density, $L$ latent heat coefficient, and $c_{\text{p}}$ specific heat capacity at constant pressure. At the surface, due to the

boundary condition, $w = 0$, $H_{\text{les,g}}$ and $E_{\text{les,g}}$ equal to zero. In LES models, $H_{\text{les}}$ and $LE_{\text{les}}$ are conventionally obtained by employing a LSM via the surface energy and water balance equations, *i.e.*,

$$R_{\text{n}} - H_{\text{les,sg}} - LE_{\text{les,sg}} - G = 0, \tag{5}$$

$$P - E_{\text{les,sg}} - I - R_0 = 0, \tag{6}$$

where $R_{\text{n}}$ is net radiation, $G$ ground heat flux, $P$ precipitation, $I$ infiltration, and $R_0$ surface runoff. The surface sub-grid

fluxes are then parameterized, e.g., using the aerodynamic resistance approach

$$H_{\text{les,sg}} = -\rho C_{\text{p}} \frac{(T_{\text{a}} - T_0)}{r_{\text{h,sg}}}, \tag{7}$$

$$LE_{\text{les,sg}} = -\rho L \beta \frac{(q_{\text{a}} - q_{\text{s}}(T_0))}{r_{\text{q,sg}}}, \tag{8}$$

where $T_{\text{a}}$ and $q_{\text{a}}$ are air temperature and specific humidity at the lowest model layer, respectively; $T_0$ is surface temperature and $q_{\text{s}}(T_0)$ saturation specific humidity at $T_0$; parameter $\beta$ can be expressed as a linear function of the topsoil moisture (e.g.

Irannejad and Shao, 1998); and $r_{\text{h,sg}}$ and $r_{\text{q,sg}}$ are aerodynamic resistances for heat and water vapor, respectively, commonly estimated using the MOST similarity functions which, however, do not apply to LES scale problems.

On the other hand, sub-grid eddy viscosity $K_{\text{m,sg}}$ and eddy diffusivity (e.g. for heat) $K_{\text{h,sg}}$ can be estimated via the LES turbulence closure scheme. For a *k-l* closure, $K_{\text{m,sg}}$ is expressed as

$$K_{\text{m,sg}} = C_k l \sqrt{e}. \tag{9}$$

where $e$ is the sub-grid turbulent kinetic energy (TKE), obtained by solving the TKE equation in the LES model, and $C_k$ is an empirical parameter of about 0.15. The mixing length $l$ is commonly set to the LES model grid resolution Δx. The sub-grid eddy diffusivity can be expressed as

$$K_{\text{h,sg}} = K_{\text{m,sg}} \text{P}_r^{-1}, \tag{10}$$

where $P_r$ is the Prandtl number, about 0.3.

The eddy diffusivity can be in turn used to estimate the aerodynamic resistance, e.g.,





$$r_{h,sg} = \int_{z_{0s}}^{z_1} K_{h,sg}^{-1}(z)\,\mathrm{d}z, \tag{11}$$

where $z_{0s}$ is the aerodynamic roughness affected by the local aerodynamic characteristics of the land surface, and $z_1$ is the height of lowest model layer. It is plausible to assume that

$$K_{h,sg}(z) = K_{h,sg}(z_1)\left(\frac{z}{z_1}\right)^n, \tag{12}$$

where $K_{h,sg}(z_1)$ is the sub-grid eddy diffusivity at $z_1$. Then, for $n = 1$, we have

$$r_{h,sg} = \frac{z_1}{K_{h,sg}(z)}, \tag{13}$$

and for other $n$ values,

$$r_{h,sg} = \frac{z_1}{(1-n)K_{h,sg}(z_1)}\left[1 - \left(\frac{z_1}{z_{0s}}\right)^{n-1}\right], \tag{14}$$

For simplicity, we assume that $r_{h,sg} = r_{q,sg}$ in Eqs. (7) and (8) which are then used to compute sub-grid surface fluxes.

## 2.2 MOST macroscopic constraint

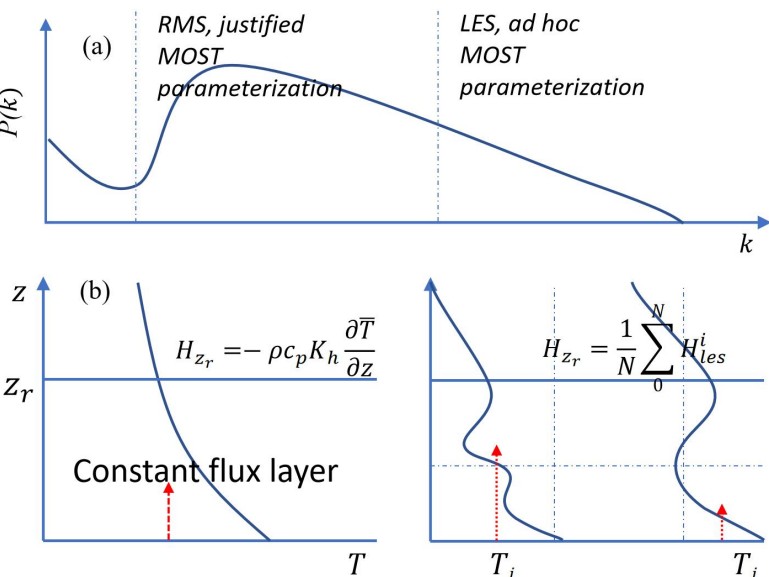

**Figure 2.** (**a**) Schematic energy spectrum of eddies, $P(k)$, as a function of wave number $k$; (**b**) Schematic profile of $T$ in Reynold mean simulation (RMS) models (left) and LES models (right).

Figure 2a shows the wavenumber ranges represented by Reynold mean simulation (RMS) and LES models, and Fig. 2b illustrates that the MOST parameterization justified for RMS models may not be justified for LES models. In the constant flux layer, for example, the typical sensible-heat flux and temperature profile relationship can be well approximated with the MOST for RMS models, but not for LES models. The use of MOST-based surface-flux parameterizations not only violates the MOST but also leads to self-inconsistencies in LES models, i.e., the MOST-based estimates of eddy viscosity and diffusivity differ from those estimated from LES sub-grid closure schemes. However, if we compute surface fluxes locally as



outlined in Section 2.1 and integrate the fluxes over a sufficiently large domain for which the MOST works well, then it is

required that

$$\frac{1}{N}\sum_1^N H^i_{\text{les}} = H_{\text{most}} = -\rho c_p K_h \frac{\partial \bar{T}}{\partial z}, \tag{15}$$

where $H^i_{\text{les}}$ is the surface sensible heat flux estimated by LES for grid cell $i$, $N$ is the total number of grid cells in the domain

and $\bar{T}$ is the average temperature over the domain. Eq. (15) is not warranted if the fluxes are simply computed as stated in

Section 2.1. Thus, a macroscopic constraint needs to be applied to the local surface flux estimates, such that Eq. (15) is

satisfied.

We use sensible heat flux for the discussion of the macroscopic constraint. Consider a LES domain that is much larger than

the scale of surface heterogeneity and corresponds to a scale on which MOST applies. Then, we expect that

$$\frac{1}{N}\sum_0^N H^i_{\text{les}} = \alpha H_{\text{most}}, \tag{16}$$

where $H_{\text{most}}$ is the constraint sensible heat flux for the LES domain and $\alpha$ is a coefficient accounting for the inaccuracy of

$H_{\text{most}}$.

For a homogeneous surface, $\alpha$ is expected to be close to one, while for other surfaces, to differ from one as the MOST

assumptions are not fully met. For a domain much larger than the scale of surface heterogeneity, we expect $\alpha$ not to differ

substantially from one. A simple correction to $H^i_{\text{les}}$ is made, such that (15) is satisfied, namely,

$$H^i_{\text{les,new}} = H^i_{\text{les}} \frac{\alpha H_{\text{most}}}{\frac{1}{N}\sum_0^N H^i_{\text{les}}}, \tag{17}$$

where $H^i_{\text{les,new}}$ is the updated surface sensible heat flux. At the surface, vertical velocity $w^i_g = 0$, and $H^i_{\text{les}}$ is entirely subgrid,

i.e.,

$$H^i_{\text{les}} = -\rho C_p K^i_{\text{h,sg}} \frac{\partial T^i_g}{\partial z} \tag{18}$$

Hence, the macroscopic constraint becomes a constraint on the LES sub-grid eddy diffusivity

$$K^i_{\text{h,sg, new}} = K^i_{\text{h,sg}} \frac{\alpha H_{\text{most}}}{\frac{1}{N}\sum_0^N H^i_{\text{les}}} \tag{19}$$

The same formulation applies to latent heat flux and momentum flux.

In practice, $H_{\text{most}}$ can be estimated by using several methods:

(1) Method 1: We use LES domain averaged surface temperature $\bar{T}_0$ and air temperature $\bar{T}_a$, to compute $H_{\text{most}}$ as

$$H_{\text{most}} = -\rho C_p \frac{(\bar{T}_a - \bar{T}_0)}{\bar{r}_h}, \tag{20}$$

where $\bar{r}_h$ is the aerodynamic resistance estimated using the MOST for the LES domain;

(2) Method 2: Suppose the LES domain consists of $J$ land use types with $\sigma_j$ being the fraction of land use type $j$. Then, $H_{\text{most}}$

can be approximated using the mosaic approach (Niu et al., 2011), using a LSM such as the Noah-MP

$$H_{\text{most}} = \sum_j \sigma_j H_j \tag{21}$$

and



$$H_j = -\rho C_p \frac{(\overline{T}_{a,j} - \overline{T}_{0,j})}{\overline{r}_{h,j}} \qquad (22)$$

where $\overline{T}_{a,j}$, $\overline{T}_{0,j}$ and $\overline{r}_{h,j}$ are mean air temperature, mean surface temperature and aerodynamic resistance for land use type $j$, respectively;

(3) Method 3: As for Method 2, but $H_j$ is calculated using the LSM driven with observed meteorological data for the different land use types.

For any given surface, as the true flux is not known, the coefficient $\alpha$ introduced in Eq. (17) as a measure of uncertainties embedded in the various flux estimates on macro scale (i.e., $H_{most}$), either parameterized or observed, is not known. However, a general understanding of $\alpha$ can be achieved through an inter-comparison of various estimates. For a relatively homogeneous surface, it is plausible to assume that measurements reasonably well represent the true macroscopic fluxes, and thus a plot of the parameterized fluxes against the observed fluxes can be interpreted as $\alpha$. For example,

$$H_{obs} = \alpha H_{most}, \qquad (23)$$

where $H_{obs}$ is sensible heat flux observation. Then, the macroscopic fluxes estimated using Method 1, 2 and 3 are compared with the observations.

For a more homogeneous surface, the idealized experiments are performed as follows to determine $\alpha$ value. Two land use types, i.e., "Mixed dryland/irrigated cropland and pasture" (hereafter MDICP) and "Urban and built-up land" (hereafter UBL)
are selected for the experiment over homogeneous surface. The model domain is 5 km × 5 km with a depth of 2.6 km. The number of vertical layer is 100 with a resolution $\Delta z$ stretching from 10 m to 40 m. The horizontal grid spacing $\Delta x = \Delta y = $ 50 m. A Rayleigh damping layer is set at 500 m from the top to damp the gravity waves. The initial horizontal wind speed, potential temperature and humidity profiles are obtained from the soundings at the Zhangye Station (39.08°N, 100.27°E, 1556.06 m ASL) at 0800 local time (LT) on August 20, 2012 (Fig. 4). The model is forced by the solar shortwave radiation
and upward longwave radiation flux observed at the Daman Station (38.78°N, 100.49°E, 1594.00 m ASL) from 0800 to 1800 LT. The initial soil temperature and soil moisture for MDICP and UBL, are represented by the observations at 0800 LT on August 20, 2012 at the Daman and Village Stations, respectively (Fig. 3b). Each case runs for 10 hours. For the simulation period, the weather was sunny and there was no influence from the weather system. The physics parameterization schemes are selected as follows: the *k-l* scheme (Deardorff, 1980; Zhang et al. 2018) is selected for the subgrid closure, and
the revised MM5 Monin-Obukhov scheme is used for the surface layer (Jimenez, 2012). Surface heat flux observations are obtained from the eddy covariance (EC) system at the Daman and Village Stations, representing the fluxes over the MDICP and UBL surfaces, respectively. The flux measurements are quality checked (Liu et al., 2016; Zhang et al., 2016) and provide an additional references for the simulations. The scatter plots (Fig. 5a-d) show the degree of dispersion between observed fluxes and those estimated using Method 1, 2, and 3, together with the fittings of the estimated $H_{most}$ to the
observations. Figure 5 shows that, for sensible heat flux, the $\alpha$ values for Method 1, 2, 3 fall between 0.91 and 0.93, 1.03 and 1.10 for MDICP and UBL, respectively. For latent heat flux, these values fall between 0.90 and 0.92, 1.09 and 1.14, for MDICP and UBL, respectively. These comparisons show that, the estimated $H_{most}$ and $LE_{most}$ by using Method 1, 2, and 3 are





generally consistent with EC measurements. This confirms our hypothesis that for a more homogeneous surfaces, all three methods for $\alpha$ are not substantially different from one.

For a somewhat heterogeneous surface, real experiment are performed to determine the $\alpha$ value as follows. A domain at Zhangye in HiWATER (Fig. 3b), which includes all three land use types MDICP, UBL and "Cropland/woodland mosaic" (hereafter CWM), is selected for the simulation. It consists of $100\times100\times100$ grid points, with $\Delta x=\Delta y=50$ m, and $\Delta z$ increases from 10 m to 40 m. The time step is 0.4 s. The initial input, and physical parameterizations are similar with the idealized experiments. The initial soil temperature and soil moisture for MDICP, UBL and CWM, are represented by the observations

at 0800 LT on August 20, 2012 at the Daman, Village and Orchard Stations, respectively. The domain averaged surface heat flux observations were obtained by a large aperture scintillometer (LAS) (Liu et al., 2016). They are used to validate the domain averaged simulated fluxes. For a somewhat heterogeneous surface, for sensible heat flux, the $\alpha$ values for Method 1, 2, 3 fall between 1.11 and 1.13; for latent heat flux, the $\alpha$ values for Method 1, 2, 3 fall between 0.90 and 0.92 (Fig. 6). The $\alpha$ values are found to be close to one. All three methods give $\alpha$ with somewhat larger scatter, but are still sufficiently close. It

is thus justified to estimate the macroscopic constraint using Method 1. The use of Method 1 is advantageous, because it provides a self-constraint using the data generated by LES and does not need additional independent information.

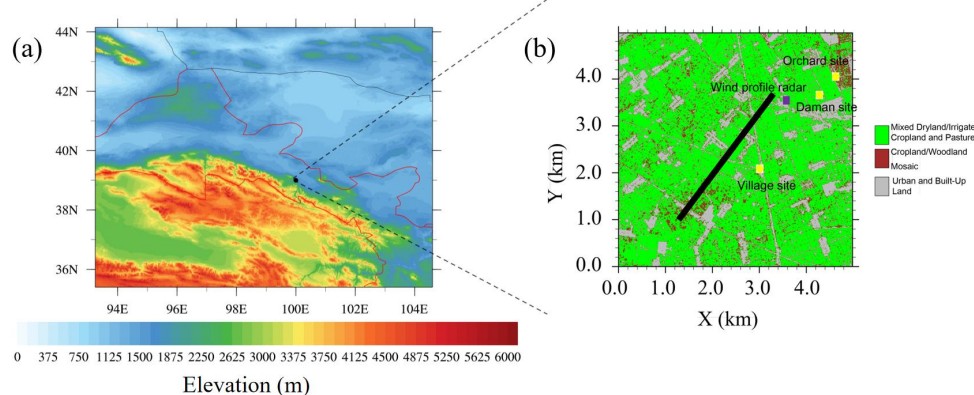

**Figure 3.** (**a**) The location of observation site and (**b**) land use map. The black solid line is the optical length of large aperture scintillometer (LAS).



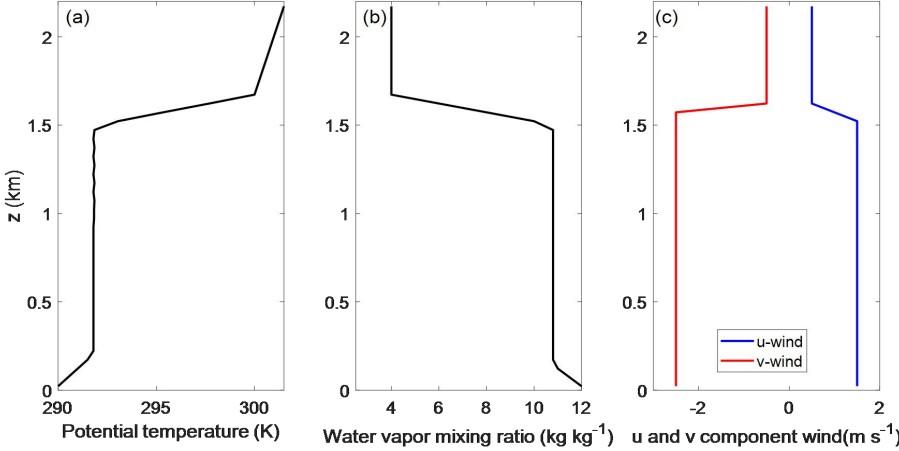

**Figure 4.** (**a**) Initial potential temperature, (**b**) water vapor mixing ratio, and (**c**) $u$, $v$ components of wind speed, derived based on the soundings at the Zhangye Station (39.08°N, 100.27°E, 1556.06 m ASL) at 0800 local time (LT) on August 20, 2012.

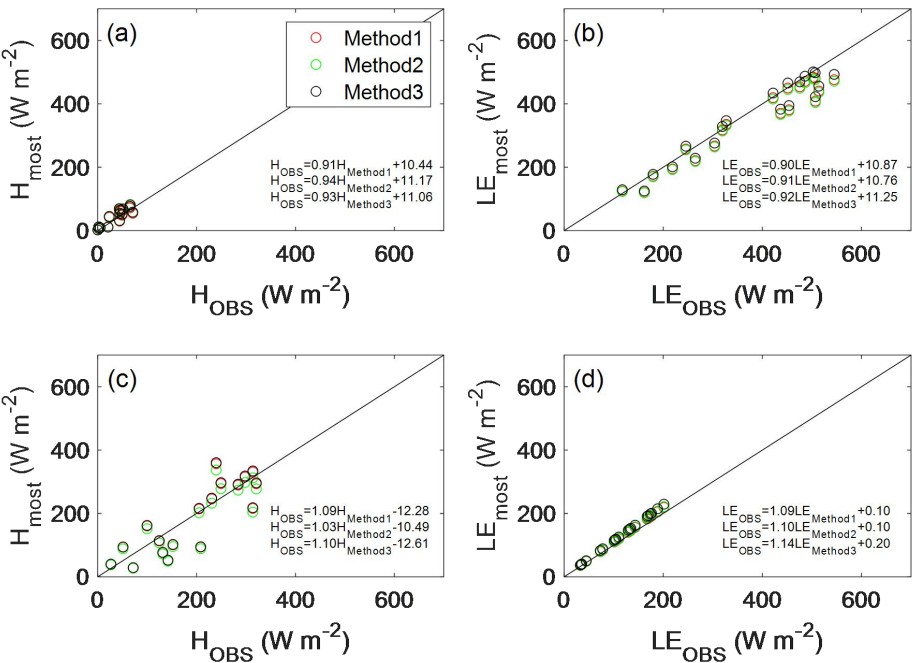

**Figure 5.** Results from simulation experiments over homogeneous surfaces. Scatter plots of (**a**) $H$ and (**b**) $LE$ estimated using Method 1, 2 and 3 (Section 2.2) against the observed fluxes for MDICP. (**c**) and (**d**) as (**a**) and (**b**), respectively, but for UBL. For the large-eddy simulations, the horizontal grid spacing is set to $\Delta x = 50$ m.





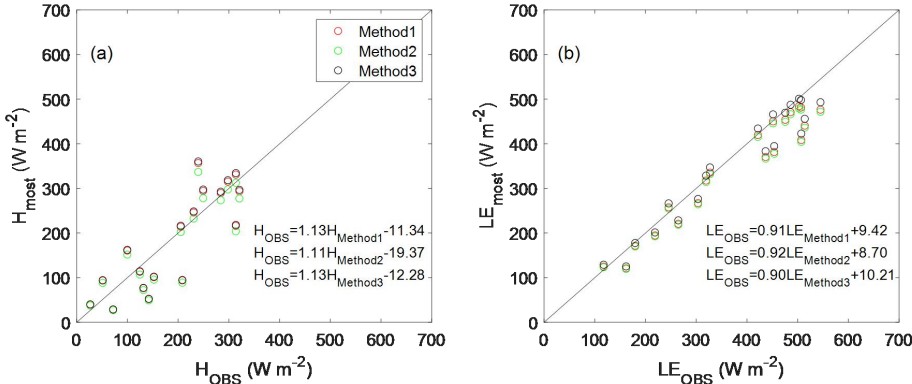

**Figure 6.** Results from real experiments. Scatter plots of (**a**) $H$ and (**b**) $LE$ estimated using Method 1, 2 and 3 (Section 2.2) against the observed fluxes. For the large-eddy simulations, the horizontal grid spacing is set to $\Delta x = 50$ m.

## 3 Real experiment setup

The real experiment setup is similar with that in section 2.2, except the surface flux scheme. The $\alpha$ values by Method 1 are used as macroscopic constraint in the MOST-r scheme in real cases. For comparison, three surface flux schemes are used, namely, one with the new scheme developed here (hereafter MOST-r), one with the Noah-MP land-surface scheme using MOST-based flux formulations for LES scheme (LES-Noah), one with the local flux calculation scheme of Shao et al. (2013) (hereafter LES-S13). In addition, the effects of different horizontal grid resolutions on the MOST-r results are investigated by changing the model grid spacing $\Delta x = \Delta y$ from 100, to 50, 25 and 10 m. The corresponding time steps are adjusted to satisfy the numerical stability conditions. The real experiments are listed in Table 1. The root mean square error (rmse) is used to measure the difference between different methods and ground measured data:

$$\text{rmse} = [M^{-1} \sum (f^{\text{obs}} - f)^2]^{1/2}, \tag{24}$$

where $M$ is number of observations, $f^{\text{obs}}$ are the fluxes measured by EC or LAS measurement system and $f$ are the fluxes calculated by different methods.

**Table 1.** Lists of real experiments.

| Case | EXP 1 | EXP 2 | EXP 3 | EXP4 |
|---|---|---|---|---|
| Scheme | MOST-r | MOST-r | MOST-r | MOST-r |
| Δx (m) | 100 | 50 | 25 | 10 |
| Case | EXP5 | EXP6 | EXP7 | EXP8 |
| Scheme | LES-S13 | LES-S13 | LES-S13 | LES-S13 |
| Δx (m) | 100 | 50 | 25 | 10 |
| Case | EXP9 | EXP10 | EXP11 | EXP12 |



| Scheme | LES-Noah | LES-Noah | LES-Noah | LES-Noah |
|---|---|---|---|---|
| Δx (m) | 100 | 50 | 25 | 10 |

# 4 Results

## 4.1 Correction of MOST-r

The results by MOST-r are compared with LES-Noah and LES-S13, where LES-Noah and LES-S13 are not corrected. To evaluate the correction of MOST-r for domain averaged $H$ and $LE$ quantitatively, rmses of domain averaged $H$ and $LE$ between different experiments and observations are calculated and shown in Table 2. The estimated $H$ and $LE$ computed by MOST-r are closer to the LAS observations than those by LES-Noah and LES-S13 methods. For example, when Δx=50 m, MOST-r has correction of 22.7 W m$^{-2}$ and 55.4 W m$^{-2}$ for $H$ and $LE$ compared with LES-Noah; MOST-r has correction of 24.0 W m$^{-2}$ and 53.7 W m$^{-2}$ for $H$ and $LE$ compared with LES-S13, respectively. In general, MOST-r has the best result, followed by LES-S13, and LES-Noah has the worst result.

**Table 2.** The rmse of domain averaged $H$ and $LE$ between different experiments and observations.

|  | Scheme | Δx = 100 m | Δx = 50 m | Δx = 25 m | Δx = 10 m |
|---|---|---|---|---|---|
| | LES-Noah | 50.8 | 52.3 | 58.3 | 63.7 |
| rmse(H) (W m$^{-2}$) | LES-S13 | 51.3 | 53.6 | 56.3 | 60.0 |
| | MOST-r | 31.2 | 29.6 | 29.3 | 28.3 |
| | LES-Noah | 81.0 | 85.9 | 92.9 | 95.8 |
| rmse(LE) (W m$^{-2}$) | LES-S13 | 81.3 | 84.2 | 89.5 | 93.1 |
| | MOST-r | 31.6 | 30.5 | 30.8 | 29.9 |

## 4.2 Profiles of the time- and domain-averaged fluxes estimated by MOST-r

Profiles of the time- and domain-averaged sensible heat fluxes $H$, $H_g$ and $H_{sg}$ and latent heat fluxes $LE$, $LE_g$ and $LE_{sg}$ estimated by LES-Noah, LES-S13, MOST-r and observations are shown in Figs. 7-8. $H$ estimated by MOST-r are generally closer to the measurements than those by LES-Noah and LES-S13. The $H$ estimated by LES-Noah, LES-S13, and MOST-r at the near surface decreases with height linearly until the inversion level (Fig. 7a-c), which is similar with that in Shao et al. (2013) (Fig. 7d-f). In the bulk of the boundary layer, $H$ is mainly caused by $H_g$, and $H_{sg}$ is negligible. Close to the surface, $H_{sg}$ dominates when the scale of turbulence is fine. In addition, $LE$ estimated by MOST-r are generally closer to the measurements than those by LES-Noah and LES-S13 (Fig. 8d-f). The near-surface flux by MOST-r varies little with height, which better satisfies the assumption of a constant flux layer. From the analysis in Section 2.2 and Eq. (15), it can be seen that the flux in the constant flux layer in RMS is different from that in LES, and the previous closure scheme of LES-Noah is inconsistent with MOST, resulting in the flux in the constant flux layer changing with height. LES-S13 has no MOST





260  constraints, so the results by LES-S13 are poorer. MOST-r remedies this problem that the $H_{sg}$ by MOST-r is larger than that by LES-S13. In short, the formulation of the MOST-r has a significant quantitative effect on the surface flux.

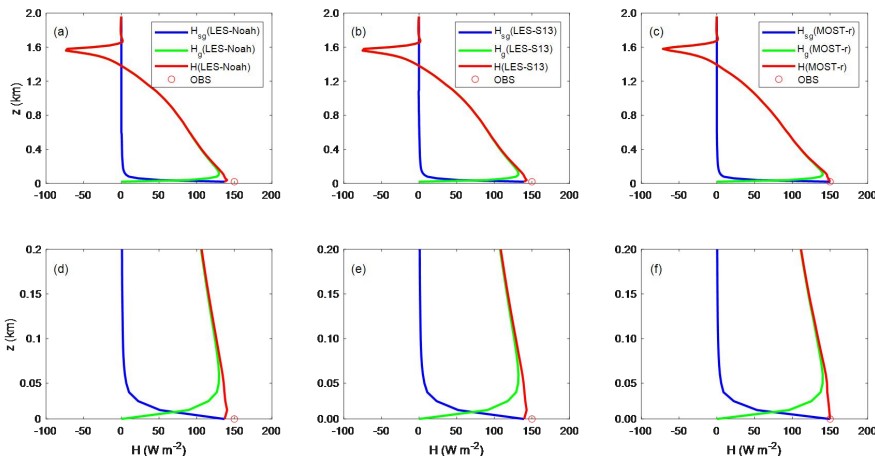

**Figure 7.** Profiles of $H$ (red line), $H_g$ (green line) and $H_{sg}$ (blue line) estimated by (**a**) LES-Noah, (**b**) LES-S13, (**c**) MOST-r averaged over the model domain and during 1200 - 1300 LT. Profiles of $H$ (red line), $H_g$ (green line) and $H_{sg}$ (blue line) estimated by (**d**) LES-Noah, (**e**)

265  LES-S13, (**f**) MOST-r for the lower 200 m. The red circle is the observation from large aperture scintillometer (LAS). For the large-eddy simulations, the horizontal grid spacing is set to $\Delta x = 50$ m.

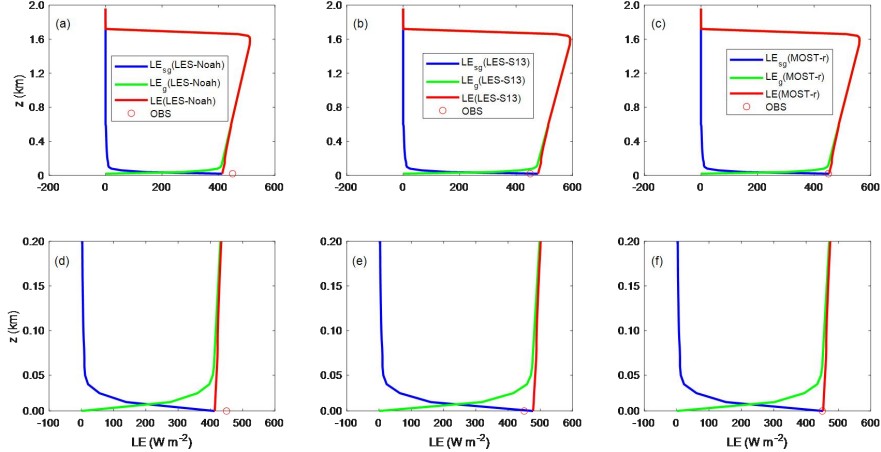

**Figure 8.** As similar with Fig. 7, but for latent heat flux.

**4.3 Patterns of surface $H$ and $LE$ estimated by MOST-r in real cases**

270  The patterns of surface $H$ and $LE$ estimated by MOST-r in real cases averaged during 1200-1300 LT are compared with those by LES-Noah and LES-S13 (Fig. 9a-c). The patterns of $H$ and $LE$ estimated by MOST-r are similar with those by LES-Noah and LES-S13, indicating that the constraint effect of MOST-r decrease over the heterogeneous surface, but MOST-r



could still constrain the pattern of heat flux unchanged. Furthermore, the patterns of $H$ and $LE$ by MOST-r are similar to the patterns of land use. For example, the maximum $H$ by MOST-r are found over the UBL areas, larger than 300 W m$^{-2}$, while the $H$ by MOST-r over the MDICP and CWM are much smaller, at around 100 W m$^{-2}$. The maximum $LE$ by MOST-r are found over the MDICP areas, in excess of 500 W m$^{-2}$, while the $LE$ by MOST-r over the UBL are much smaller, at around 200 W m$^{-2}$. In short, the surface heat flux is influenced by the pattern of land use.

The grid point results at around observation points in the LES domain are bilinearly interpolated to the observation points. Over the MDICP and UBL, the estimated $H$ and $LE$ computed by MOST-r are closer to the EC observations than those by LES-Noah and LES-S13 methods (Fig. 10). The estimates of net radiation flux ($R_{net}$), surface skin temperature ($T_s$), ground heat flux ($G$) in real cases are compared with the measurements (Fig. 11). Over MDICP and UBL, the estimated $R_{net}$ by MOST-r are closer to the observations than those by LES-Noah and LES-S13. For example, the simulated $R_{net}$ by MOST-r is up to 50 W m$^{-2}$ higher than the observed values, while $R_{net}$ by LES-S13 is more than 100 W m$^{-2}$ higher than the observed values. Furthermore, the $R_{net}$ over UBL by all methods is smaller than that over MDICP. The agreement of $T_s$ between MOST-r and the observations is evidently better than that by LES-Noah and LES-S13 over MDICP and UBL. $T_s$ is overestimated by LES-Noah and LES-S13 up to 3 °C higher than that observed. The $T_s$ over UBL by all methods is larger than that over MDICP. The $G$ estimated by MOST-r evidently matches with the observations and is better than that by LES-Noah and LES-S13 over MDICP and UBL, which is overestimated by LES-Noah up to 30 W m$^{-2}$ higher than that observed over MDICP. The above results indicate that MOST-r has better results than those by LES-Noah and LES-S13.

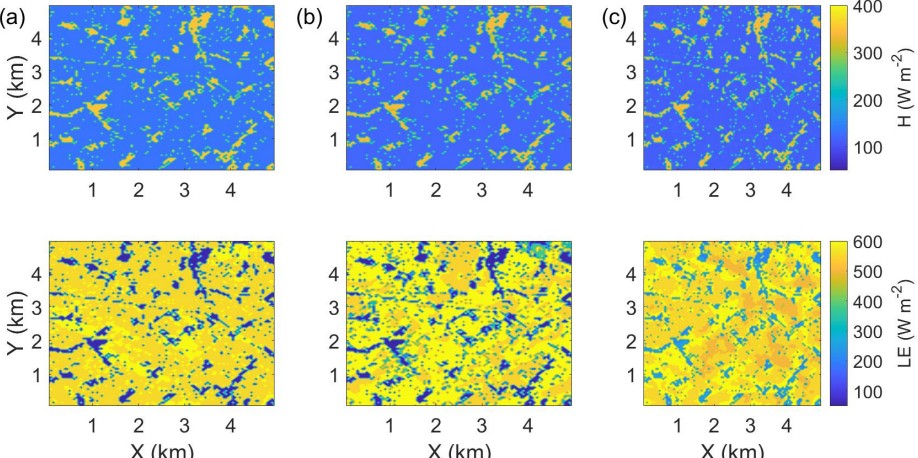

**Figure 9.** Patterns of surface sensible heat flux ($H$) and latent heat flux ($LE$) estimated by (**a**) LES-Noah, (**b**) LES-S13, (**c**) MOST-r averaged during 1200 - 1300 LT. For the large-eddy simulations, the horizontal grid spacing is set to $\Delta x = 50$ m.



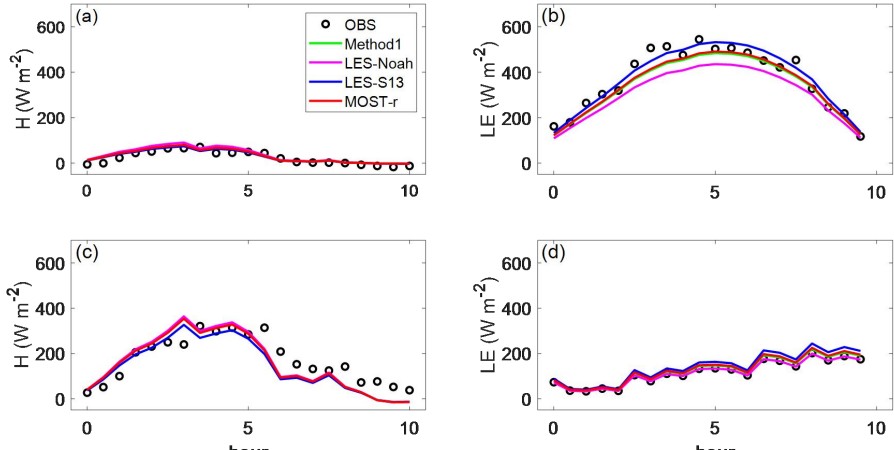

**Figure 10.** Time series of [(**a**) $H$, (**b**) $LE$] over MDICP and [(**c**) $H$, (**d**) $LE$] over UBL by observation, Method 1, LES-Noah, LES-S13, and MOST-r method with $\Delta x = 50$ m.

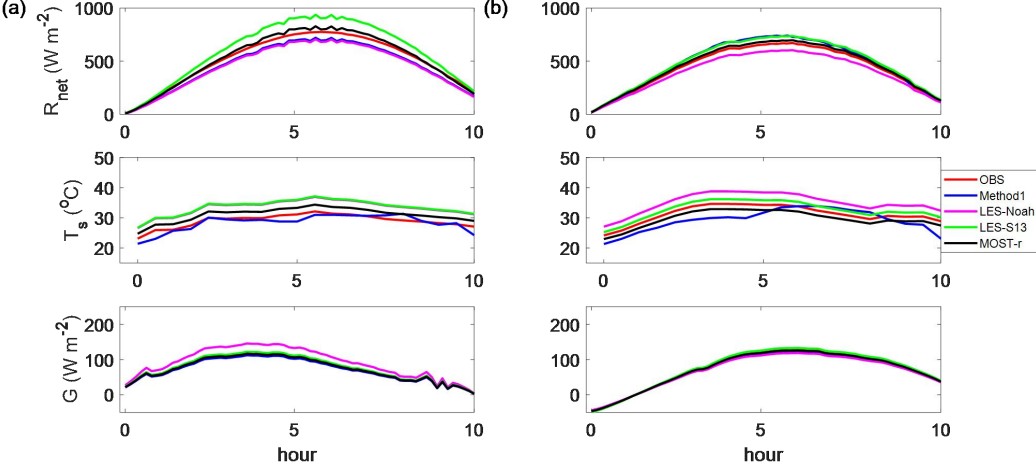

**Figure 11.** Net radiation flux ($R_{net}$), surface temperature ($T_s$), ground heat flux ($G$) estimated by Method1, LES-Noah, LES-S13, MOST-r and observations over (**a**) MDICP, (**b**) UBL surface averaged over the model domain.

## 4.4 The effect of horizontal grid spacing on the correction by MOST-r

To investigate the effect of horizontal grid spacing $\Delta x$ on the correction by MOST-r, several groups of sensitivity experiments about horizontal grid spacing $\Delta x = \Delta y = 100$ m, $50$ m, $25$ m, $10$ m are carried out. The results are shown in Figs. 12 and Table 2. The results show that the correction of $H$ and $LE$ by MOST-r keep stable with the decrease of the horizontal grid spacing (Fig. 12a-b). Surface $H_{les}$ and $LE_{les}$ estimated by LES-S13 decreases with the decrease of the $\Delta x$, indicating that $\Delta x$ has large effect on surface $H_{les}$ and $LE_{les}$ by LES-S13. However, the $H_{most}$ and $LE_{most}$ estimated by Method 1 and 2 remain almost unchanged no matter how $\Delta x$ changes. These results are similar with those in Shin and Hong (2013),





where the domain averaged sub-grid fluxes decrease with the decrease of $\Delta x$, while the resolved fluxes increase. Therefore, the results by MOST-r scheme keep stable with the change of horizontal grid spacing, similar with the results of Method 1 and 2.

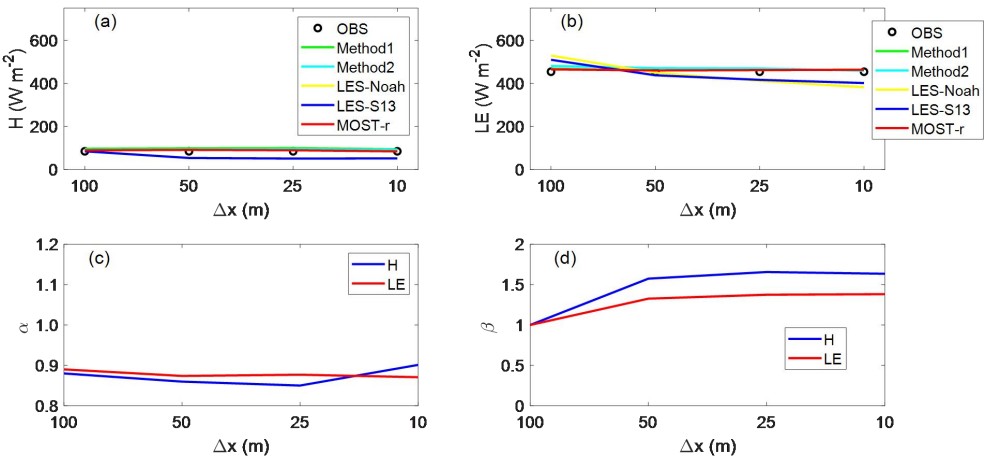


**Figure 12.** (**a**) $H$ and (**b**) $LE$ by observation, Method1, Method2, LES-Noah, LES-S13 and MOST-r averaged over the model domain and during 1200 - 1300 LT with different horizontal grid spacing ($\Delta x = 100, 50, 25, 10$ m). Variations of (**c**) $\alpha$ and (**d**) $\beta$ for $H$ and $LE$ with different horizontal grid spacing ($\Delta x = 100, 50, 25, 10$m) averaged over the model domain and during 1200 - 1300 LT.

**4.5 The effect of horizontal grid spacing on the $\alpha$**

The variations of $\alpha$ with horizontal grid spacing $\Delta x$ are shown in Fig. 12c. The $\alpha$ values for $H$ and $LE$ between observation and Method 1 remain unchanged with the change of $\Delta x$. For example, $\alpha$ values for $H$ are distributed near 0.93, and $\alpha$ values for $LE$ are distributed near 0.92 from $\Delta x = 100$m to $\Delta x = 10$m, respectively (Fig. 12c). In short, the $\alpha$ values for $H$ and $LE$ do not change with horizontal grid spacing.

To investigate the influence of $\Delta x$ on the macroscopic constraint of MOST, the right hand of Eq. (17) is defined as $\beta$:

$$\beta = \frac{\alpha H_{\text{most}}}{\frac{1}{N} \Sigma_0^N H_{\text{les}}^i} \qquad (25)$$

The results of $\beta$ are shown in the Fig. 12d. The value of $\beta$ increases with the decrease of $\Delta x$. For example, $\beta$ for $H$ reaches from 1.00 at $\Delta x = 100$ m to 1.68 at $\Delta x = 25$m, indicating that the macroscopic constraint of MOST on the LES increases with the decrease of horizontal grid spacing. In addition, $\beta$ for $LE$ is smaller than that for $H$, indicating that the macroscopic constraint of MOST on $H$ is larger than that on $LE$ (Fig. 12d).



## 5 Conclusion


When the MOST applies to the LES to estimate the surface heat fluxes for a scale at which MOST is expected to work well, there is a discrepancy between the LES and the mesoscale or large scale model. A new parameterization scheme (MOST-r) to calculate surface heat fluxes in LES is developed by invoking the macroscopic constraint using MOST, making the turbulent heat flux in LES close to results from the model based on MOST. The coefficient of $\alpha$ are used to character the

uncertainty of the surface heat flux by MOST. The optimal solution of $\alpha$ is determined by Method 1, 2 and 3 in section 2.2. $\alpha$ is close to 1 for homogeneous case, and $\alpha$ is not too different from 1 even for heterogeneous case.

The Method 1 provides a self-constraint on the LES-simulated fluxes, which are used in the MOST-r scheme to carry out the real-case experiment over oasis surface in Northwestern China. The Noah-MP land surface scheme coupled with LES (LES-Noah) and Shao et al. (2013) land surface scheme coupled with LES (LES-S13) are used to compare with this new scheme.

MOST-r has the best result, followed by LES-S13, and LES-Noah has the worst result. The constraint of MOST-r decreases over the heterogeneous surface, but MOST-r could constrain the pattern of heat flux unchanged. In addition, the estimated net radiation flux, ground heat flux and surface skin temperature by MOST-r are generally consistent with the measurements. The sensitivity experiments of horizontal resolution demonstrate that the $\alpha$ values vary little with horizontal resolution. The surface heat flux estimated by MOST-r keep stable, but they estimated by LES-S13 change with horizontal grid spacing. The

formulation of the MOST-r has a significant quantitative effect on the surface flux.

To broaden the applicability of MOST-r scheme in the future, it is necessary to apply MOST-r scheme to more real-case simulations, including that use the reanalysis data to drive the model based on MOST as the constraint to correct the surface heat flux in LES.

Code availability. The source code used in this study is the WRF version 4.3 in the LES mode. WRF model can be downloaded at https://www2.mmm.ucar.edu/wrf/users/download/get_sources.html (WRF Users page). The MOST-r scheme code can be accessed by contacting Bangjun Cao (caobj1989@163.com).

Data availability. The data presented in the paper can be accessed by contacting Bangjun Cao (caobj1989@163.com).

Author contributions. YP and BJ had the original idea of the study. BJ and Xin Yin conducted model simulation. BJ

interpreted the data and plotted the figures. BJ wrote the article with contributions from all the co-authors. Project administration, Xianyu; funding acquisition, Xianyu; review and editing, YP and SF.

Competing interests. The contact author has declared that neither they nor their co-authors have any competing interests.

Acknowledgement. Thanks to the Heihe Watershed Allied Telemetry Experimental Research, for the field observation data.





Financial support. This research has been supported by the National Natural Science Foundation of China (grant nos. 41975130, 42175174) and by the Second Tibetan Plateau Scientific Expedition and Research Program (STEP) (grant nos. 2019QZKK0102).

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
