# Peer review of "Large-Eddy Simulation of Turbulent Flux Patterns over Oasis Surface"

_EGUsphere, 2023_

## Referee Comment (RC1)

Comments on "Large-Eddy Simulation of Turbulent Flux Patterns over Oasis Surface"

The manuscript "Large-Eddy Simulation of Turbulent Flux Patterns over Oasis Surface" by Cao et al. submitted to the Atmospheric Chemistry and Physics proposes a method to improve the simulation of surface fluxes for large-eddy simulations (LESs). The premise of the study is the long-standing challenge for wall-modeled LESs to parameterize surface fluxes. Current LESs mostly rely on Monin-Obukhov similarity theory (MOST) for surface fluxes, while it is known that MOST applies to the ensemble averaged rather than instantaneous fluxes. The authors therefore apply MOST to the horizontally averaged fluxes, and use that to scale the fluxes predicted by the LES closure at the individual grid points to achieve agreement with MO at the LES domain scale. While the proposed method is of practical value, I am afraid that I must reject the manuscript in its current form. Please see my major comments for details. Meanwhile, please improve the English writing. I gave up looking for typos and grammatic errors at line 61, so please, make sure all listed authors proofread the manuscript carefully.

**Major Comments**

1. I found the physics behind the proposed method questionable. For that, I must draw the authors' attention to the two-part eddy viscosity model proposed by Sullivan et al. (1994). I think the authors are essentially pursing the same idea, to obtain agreement with MOST on the scale at which the theory is developed for. Achieving this through linear scaling of the LES fluxes with Eq. 17 gets you the right result (i.e., agreement with MO for the domain-averaged fluxes), but is not fully based on the physics of turbulence closures. I strongly encourage the authors to read Sullivan et al. (1994) carefully, and find that the LES closures are developed for isotropic turbulence, and near the wall, you essentially need a RANS model that encompasses all turbulence scales. Simply scaling up/down the LES closure to produce the RANS results are not physically-founded. For the purpose of achieving MOST for the domain averaged fluxes, the two-part eddy viscosity model is more physically meaningful, as it acknowledges the shift between LES to RANS-modeling as the surface is approached.

2. The method is not well-described with many key details missing, assumptions unaccounted for, and possible errors in the equations. For example,

- Eqs 3 and 4, it is best to formulate the turbulent fluxes with turbulent perturbations.

- Eqs. 5 and 6, these are steady state budgets. I don't think LSM assume steady state energy balance, they predict changes of surface temperature and humidity.

- Please double check Eq. 13, I think it should be z1/K(z1)*ln(z1/z0).

- Reynold mean simulation (RMS) and k-l model, I think they are most often referred to as RANS simulation and TKE-1.5 models.

- At which vertical levels (or up to what height) do the authors apply the proposed MO-correction (i.e., Eq. 17), as this is clearly a surface layer correction.

- I don't think it is necessary to introduce Eqs. 18 and 19, applying correction to the fluxes as in Eq. 17 should be sufficient.

- I think the proposed method for forcing MOST for the horizontally averaged fluxes may not apply to heterogenous surfaces despite the claims of the authors. For that to apply, the authors may need to go into the inertial sublayer (which may very well lie above the first few grid points above the wall). If this is the case, how would the authors deal with those grid points that lie within the viscous sublayer where surface heterogeneity matters.

3. The purposes of the tests to determine $\alpha$ is not clear. It seems like the authors are simply testing the validity of MOST, or how good a specific set of MO similarity functions are in predicting the surface fluxes at the measurement site. They use fluxes obtained from eddy correlation measurements as benchmark to test the performance of the MOST-parameterized fluxes. For this purpose, I don't think the authors need LES, a single column model would do.

4. The real case evolution is not targeted at the improvements the method could potential lead to. I don't think mean profiles and time series of sensible and latent heat fluxes are useful for the purpose of evaluation. If the authors ready force MOST at the domain scale, the domain averaged fluxes would be in good agreement. You do not even need LES for that, a single column model would give you similar results. The authors should focus on turbulence characteristics instead of mean profiles. The spectra might be a good place to start. The horizontal distribution of fluxes in Fig. 9 offers some insight, but it is hard to tell

the differences (also please make sure all subplots have the same sizes for ease of comparison).

**Minor Comments**

1. Line 11, often violate the …
2. Line 12, better turbulence closure than model closure
3. Line 13-14. … LES models. It computes …
4. Line 14, here and elsewhere, "closure" rather than "closure scheme"
5. Line 20, how would a constraint increase with resolution? Please rephrase for clarity
6. Line 22, exchange
7. Line 26, "to be" rather than "is"
8. Line 41, here and elsewhere, Deardorff's model is commonly referred to as the 1.5-order TKE scheme, rather than $k - l$ scheme. The $k - l$ scheme usually refers to two equation models, which include a separate prognostic equation for the length scale $l$ (specifically $kl$), see for Holt and Raman (1988) for a review.
9. Line 43, better prescribed then pre-specified.
10. Lines 52-53, "Second, how to estimate surface fluxes in LES models is an unsolved problem". Not sure what you are trying to say here, you might need to be more specific. How is this related to "scale differences"?
11. Line 56, based on
12. Lines 58-60, please refer to the work of Redelsperger et al. (2001) that addresses exactly point 3.
13. Line 61, nighttime, one word, as for daytime.

**Reference:**

Holt, T., and S. Raman. 1988. "A Review and Comparative Evaluation of Multilevel Boundary Layer Parameterizations for First-Order and Turbulent Kinetic Energy Closure Schemes." *Reviews of Geophysics* 26 (4): 761–80. https://doi.org/10.1029/RG026i004p00761.

Redelsperger, J. L., F. Mahe, and P. Carlotti. 2001. "A Simple and General Subgrid Model Suitable Both for Surface Layer and Free-Stream Turbulence." *Boundary-Layer Meteorology* 101 (3): 375–408. https://doi.org/10.1023/A:1019206001292.

Sullivan, P., J. C. Mcwilliams, and C. Moeng. 1994. "A Subgrid-Scale Model for Large-Eddy Simulation of Planetary Boundary-Layer Flows." *Boundary-Layer Meteorology* 71 (3): 247–76. https://doi.org/10.1007/BF00713741.

---

## Author Comment (AC1)

**Replies to All Referees and Editors**

**Manuscript number**: egusphere-2023-148

**Author(s)**: Bangjun Cao, Yaping Shao, Xianyu Yang, Xin Yin, Shaofeng Liu

**Title**: Large-Eddy Simulation of Turbulent Flux Patterns over Oasis Surface

**September 2023**

**I.   Response to Comments of Reviewer #1**

**[General comments]**

The manuscript "Large-Eddy Simulation of Turbulent Flux Patterns over Oasis Surface" by Cao et al. submitted to the Atmospheric Chemistry and Physics proposes a method to improve the simulation of surface fluxes for large-eddy simulations (LESs). The premise of the study is the long-standing challenge for wall-modeled LESs to parameterize surface fluxes. Current LESs mostly rely on Monin-Obukhov similarity theory (MOST) for surface fluxes, while it is known that MOST applies to the ensemble averaged rather than instantaneous fluxes. The authors therefore apply MOST to the horizontally averaged fluxes, and use that to scale the fluxes predicted by the LES closure at the individual grid points to achieve agreement with MO at the LES domain scale. While the proposed method is of practical value, I am afraid that I must reject the manuscript in its current form. Please see my major comments for details. Meanwhile, please improve the English writing. I gave up looking for typos and grammatical errors at line 61, so please, make sure all listed authors proofread the manuscript carefully.

Response: We kindly thank the anonymous reviewer for your appreciable comments that help us much in improving the quality of the manuscript. We have carefully considered your suggestions and have made the necessary revisions to enhance the writing style and clarity of the manuscript. We have taken steps to improve the readability and overall understanding of the content, making sure that our research is effectively communicated to our readers. All the comments have been seriously considered, and the manuscript is revised accordingly to meet the publication standard in Atmospheric Chemistry and Physics.

**[Major Comments]**

1. I found the physics behind the proposed method questionable. For that, I must draw the authors' attention to the two-part eddy viscosity model proposed by Sullivan et al. (1994). I think the authors are essentially pursing the same idea, to obtain agreement with MOST on the scale at which the theory is developed for. Achieving this through linear scaling of the LES fluxes with Eq. 17 gets you the right result (i.e., agreement with MO for the domain-averaged fluxes), but is not fully based on the physics of turbulence closures. I strongly encourage the authors to read

Sullivan et al. (1994) carefully, and find that the LES closures are developed for isotropic turbulence, and near the wall, you essentially need a RANS model that encompasses all turbulence scales. Simply scaling up/down the LES closure to produce the RANS results are not physically-founded. For the purpose of achieving MOST for the domain averaged fluxes, the two-part eddy viscosity model is more physically meaningful, as it acknowledges the shift between LES to RANS-modeling as the surface is approached.

Response: We greatly appreciate your insightful comment on our manuscript. The problem of inconsistency between subgrid closure and the MOST in LES models has been studied by Sullivan et al. (1994). Instead of abandoning the use of the MOST, the latter authors proposed a two-part [a turbulent (large eddy) part and a mean-flow part] SGS eddy-viscosity model, to achieve the better agreement between LES and MOST similarity forms in the surface layer. In their model, the usual SGS turbulent kinetic energy formulation for the SGS eddy viscosity is preserved, but a contribution from the mean flow is explicitly included, and the contributions from the turbulent part are reduced near the surface. Sullivan et al. (1994) reported that their model yielded increased fluctuation amplitudes near the surface and better correspondence with similarity forms in the surface layer. The review of Sullivan et al. (1994) has been added to the revised manuscript. Please see the details in Lines 46-52.

We acknowledge the significance of further exploring the intricacies of the two-part eddy viscosity model, which will substantially augment the rigor and comprehensiveness of our study. After careful consideration, we have decided to take your advice into account for future work. While the two-part eddy viscosity model provides an interesting approach to aligning LES SGS with the MOST, it did not explicitly provide a solution for surface flux estimates in LES models. A questionable assumption of their model is the reduced contribution of the turbulent (large eddy) part and increased contribution of the mean part to the subgrid eddy viscosity near the surface, because this assumption reduces the importance of large eddies which may arise due to surface heterogeneity and thus does not preserve the flux patterns, although the mean values of the flux (in their case, surface shear stress) may be preserved. The revised manuscript has added a discussion of limitations and outlook employing the work by Sullivan et al. (1994). Please see the details in Lines 330-337.

In addition, we strengthen more physical insights beyond a purely mathematical correction in the revised manuscript for our proposed method as follows:

Estimating surface fluxes for LES models is usually also based on the MOST. However, the near-surface diffusivity and viscosity estimated by the MOST-based schemes often diverge from those derived from LES subgrid closures, causing inconsistencies between them (Redelsperger et al., 2001).

To deal with the inconsistency problem, a strategy is proposed by Shao et al. (2013) to estimate surface flux based on the subgrid turbulence intensity, parameterized by the subgrid turbulence closure, for LES models. This strategy ensures that the surface flux estimates and subgrid closure are on a consistent physical basis. However, this strategy requires an extrapolation of eddy diffusivity and viscosity to the surface and thus local surface parameters (e.g., local roughness length), and it is not clear whether the surface fluxes estimated this way satisfy the MOST on the scale for which the theory works well.

[Figure]

**Figure 1. (a)** Schematic energy spectrum of eddies, $P(k)$, as a function of wave number $k$; **(b)** Schematic profile of $T$ in Reynold averaged Navier-stokes (RANS) models (left) and LES models (right). $z_r$ is the height of the constant flux layer.

Figure 1a shows the wavenumber ranges represented by RANS and LES models, while Fig. 1b visually demonstrates that the MOST parameterization, suitable for RANS models, may not hold true for LES models. In the constant flux layer, such as

the sensible-heat flux and temperature profiles, the relationship can be well approximated with the MOST for RANS models, but deviates when applied to LES models. Utilizing MOST-based surface-flux parameterizations contradicts the MOST and introduces internal inconsistencies within LES models, manifesting as disparities between MOST-based estimations of eddy viscosity and diffusivity and those derived from LES sub-grid closure. However, if we compute surface fluxes locally, as outlined in Shao et al. (2013), and integrate the fluxes over a sufficiently large domain for which the MOST works well, then it is required that

$$\frac{1}{N}\Sigma_1^N H_{\mathrm{les}}^i = H_{\mathrm{most}} = -\rho c_p K_h \frac{\partial \overline{T}}{\partial z}, \tag{1}$$

where $H_{\mathrm{les}}^i$ is the surface sensible heat flux estimated by LES for grid cell $i$, $N$ is the total number of grid cells in the domain and $\overline{T}$ is the average temperature over the domain. Eq. (1) is not warranted if the fluxes are simply computed, as stated in Shao et al. (2013). Thus, a macroscopic constraint needs to be applied to the local surface flux estimates to ensure adherence to Eq. (1).

For the purpose of achieving MOST for domain averaged surface fluxes estimated by Shao et al. (2013), this study presents an innovative approach for surface flux calculation for LES models. This approach comprises two components. Initially, it calculates LES subgrid fluxes using eddy viscosity and diffusivity estimates derived from LES closure models without invoking the MOST while considering local turbulence characteristics. Second, it employs a macroscopic constraint to ensure that fluxes averaged over the LES domain, which corresponds to scales suitable for the MOST application, align with the principles of MOST (Fig. 2).

Although simply scaling up/down the LES closure to produce the RANS results, our new scheme (MOST-r) impressively preserves the overall heat flux pattern. This adaptability underscores the robustness of our new method's constraints, which can accommodate diverse surface characteristics while maintaining the fundamental heat flux pattern. The two-part eddy viscosity model proposed by Sullivan et al. (1994) provided important insight into achieving alignment of subgrid closure with MOST cross different spatial scales. But as far as flux estimates in LES models are concerned, the use of eddy viscosity and diffusivity derived from the LES turbulent closure accounts for the heterogeneities on scales larger than the LES model resolution. Hence, the MOST-r scheme already integrated the basic ideas of the two-part eddy viscosity model of Sullivan et al. (1994), without invoking the assumption of reduced

turbulent contribution to subgrid eddy viscosity and diffusivity. This enables the MOST-r scheme to better model the surface flux patterns. Instead of attempting to derive a general scale-invariant scheme for flux estimates, we provided a scheme that is both simple and effective for LES models.

These contents have been added to the revised manuscript. Please see details in Lines 31-152 in the revised manuscript.

[Figure]

**Figure 2.** Schematic diagram of the MOST-r scheme.

2. The method is not well-described with many key details missing, assumptions unaccounted for, and possible errors in the equations. For example, Eqs 3 and 4, it is best to formulate the turbulent fluxes with turbulent perturbations.

Response: Done. Please see details in Lines 83-84 in the revised manuscript.

Eqs. 5 and 6, these are steady state budgets. I don't think LSM assume steady state energy balance, they predict changes of surface temperature and humidity.

Response: Corrected. Please see details in Lines 88-89 in the revised manuscript.

Please double check Eq. 13, I think it should be z1/K(z1)*ln(z1/z0).

Response: Corrected. Please see details in Line 112 in the revised manuscript.

Reynold mean simulation (RMS) and k-l model, I think they are most often referred to as RANS simulation and TKE-1.5 models.

Response: Corrected.

At which vertical levels (or up to what height) do the authors apply the proposed MO- correction (i.e., Eq. 17), as this is clearly a surface layer correction.

Response: We apply the proposed MOST correction up to a height of 50 m. This correction indeed pertains to the surface layer. Please see details in Lines 138-139 in the revised manuscript.

I don't think it is necessary to introduce Eqs. 18 and 19, applying correction to the fluxes as in Eq. 17 should be sufficient.

Response: We aim to enhance the calculation of the LES subgrid eddy viscosity and diffusivity through constraint of MOST, subsequently attaining the macroscopic constraint on flux. The amelioration of latent heat flux and momentum flux is also realized by refining the subgrid eddy viscosity and diffusivity.

I think the proposed method for forcing MOST for the horizontally averaged fluxes may not apply to heterogeneous surfaces despite the claims of the authors. For that to apply, the authors may need to go into the inertial sublayer (which may very well lie above the first few grid points above the wall). If this is the case, how would the authors deal with those grid points that lie within the viscous sublayer where surface heterogeneity matters.

Response: We appreciate your insightful comment on the applicability of our proposed method for enforcing MOST on horizontally averaged fluxes, particularly over heterogeneous surfaces. Currently, $z_{0s}$ from different land use types were obtained from a lookup table. Furthermore, there is no specialized treatment for grid points within the viscous sublayer where surface heterogeneity plays a significant role.

We acknowledge the complexities that surface heterogeneity can introduce. We agree that addressing this issue may require delving into the inertial sublayer, which could extend beyond the initial grid points near the surface. Should we pursue this direction in the future, we will carefully devise a strategy for handling grid points within the viscous sublayer, where surface heterogeneity exerts significant influence.

3. The purposes of the tests to determine α is not clear. It seems like the authors are simply testing the validity of MOST, or how good a specific set of MO similarity functions are in predicting the surface fluxes at the measurement site. They use fluxes obtained from eddy correlation measurements as benchmark to test the

performance of the MOST- parameterized fluxes. For this purpose, I don't think the authors need LES, a single column model would do.

Response: We appreciate your feedback and apologize for any lack of clarity in our previous description. $\alpha_j$ represents the efficiency factor of $H_{most}$ for land use type $j$. It's important to highlight that the using the LES domain is necessary to obtain essential parameters such as mean profiles of air temperature, mean surface temperature, and other relevant parameters. Additionally, the LES enables us to investigate the spatial pattern of flux, a capability that a Single Column Model (SCM) lacks. To improve clarity, we have revised the description in Section 2.2 of our manuscript. Please refer to the revised manuscript in Lines 121-152 for a more detailed explanation.

For further clarification on the methodology used to derive these $\alpha$ values, we have provided comprehensive details in the Supplementary Material section of our manuscript.

4. The real case evolution is not targeted at the improvements the method could potential lead to. I don't think mean profiles and time series of sensible and latent heat fluxes are useful for the purpose of evaluation. If the authors ready force MOST at the domain scale, the domain averaged fluxes would be in good agreement. You do not even need LES for that, a single column model would give you similar results.

Response: We appreciate your comment and apologize for any lack of clarity in our previous description. It's important to highlight that using the LES domain is necessary to obtain essential parameters such as mean profiles of air temperature, mean surface temperature, and other relevant parameters. Additionally, the LES enables us to investigate the spatial patterns of flux, a capability that a Single Column Model (SCM) lacks. To improve clarity, we have revised the description in Section 2.2 of our manuscript. Please refer to the revised manuscript in Lines 121-152 for a more detailed explanation.

The authors should focus on turbulence characteristics instead of mean profiles. The spectra might be a good place to start. The horizontal distribution of fluxes in Fig. 9 offers some insight, but it is hard to tell the differences (also please make sure all subplots have the same sizes for ease of comparison).

Response: We greatly appreciate your insightful suggestion on our manuscript. We have revised and ensured all subplots have the same sizes for ease of comparison in Fig. 7 in the revised manuscript. Your suggestions for focusing on turbulence characteristics and exploring spectra are valuable and align with the essence of our research.

After careful consideration, we have decided to take your advice into account for future work. We recognize the importance of delving deeper into turbulence characteristics and spectra, and this will significantly enhance the quality and depth of our research. We sincerely appreciate your constructive comment, which will undoubtedly improve our work in subsequent studies. Your insights have been instrumental in guiding our research in a more focused and effective direction.

**[Minor Comments]**

1. Line 11, often violate the …

Response: Corrected.

2. Line 12, better turbulence closure than model closure

Response: Corrected.

3. Line 13-14. … LES models. It computes …

Response: Corrected.

4. Line 14, here and elsewhere, "closure" rather than "closure scheme"

Response: Corrected.

5. Line 20, how would a constraint increase with resolution? Please rephrase for clarity

Response: We have rephrased this point in the manuscript for clarity. Our sensitivity experiments, focusing on horizontal resolution, underscore the robustness of our scheme, as it maintains its corrective efficacy despite changes in horizontal grid spacing. We find that the macroscopic constraint imposed by MOST on LES-estimated fluxes strengthens as the horizontal grid spacing decreases, with a more pronounced influence on sensible than latent heat fluxes. These findings collectively

highlight the promise and adaptability of our scheme for enhancing surface flux estimations in LES models. Please see details in the revised manuscript in Lines 18-23.

6. Line 22, exchange

Response: Corrected.

7. Line 26, "to be" rather than "is"

Response: Corrected.

8. Line 41, here and elsewhere, Deardorff's model is commonly referred to as the 1.5-order TKE scheme, rather than $k - l$ scheme. The $k - l$ scheme usually refers to two equation models, which include a separate prognostic equation for the length scale $l$ (specifically $kl$), see for Holt and Raman (1988) for a review.

Response: Corrected. The reference of Holt and Raman (1988) has been added to the revised manuscript.

8. Line 43, better prescribed then pre-specified.

Response: Corrected.

10. Lines 52-53, " Second, how to estimate surface fluxes in LES models is an unsolved problem" . Not sure what you are trying to say here, you might need to be more specific. How is this related to "scale differences"?

Response: Thank you for your comment. We apologize for any confusion. We have rephrased this point in the manuscript for clarity. Please see details in the revised manuscript in Lines 38-40.

11. Line 56, based on

Response: Corrected.

12.Lines 58-60, please refer to the work of Redelsperger et al. (2001) that addresses exactly point 3.

Response: Corrected. The reference of Redelsperger et al. (2001) has been added to the revised manuscript.

13. Line 61, nighttime, one word, as for daytime.

Response: Corrected.

**Reference:**

Holt, T., and S. Raman. 1988. "A Review and Comparative Evaluation of Multilevel Boundary Layer Parameterizations for First-Order and Turbulent Kinetic Energy Closure Schemes." Reviews of Geophysics 26 (4): 761–80.https://doi.org/10.1029/RG026i004p00761.

Redelsperger, J. L., F. Mahe, and P. Carlotti. 2001. "A Simple and General Subgrid ModelSuitable Both for Surface Layer and Free-Stream Turbulence." Boundary-Layer Meteorology 101 (3): 375–408. https://doi.org/10.1023/A:1019206001292.

Sullivan, P., J. C. Mcwilliams, and C. Moeng. 1994. "A Subgrid-Scale Model for Large-Eddy Simulation of Planetary Boundary-Layer Flows." Boundary-Layer Meteorology 71(3): 247–76. https://doi.org/10.1007/BF00713741.

**II. Response to Comments of Reviewer #2**

**[General comments]**

The manuscript titled "Large-Eddy Simulation of Turbulent Flux Patterns over Oasis Surface" by Cao et al. discusses the issue of inconsistencies in flux calculations between Large-Eddy Simulation (LES) model closures and the Monin-Obukhov Similarity Theory (MOST) parameterization. The authors propose a revised scheme for estimating turbulent fluxes in LES models, introducing a macroscopic constraint based on the MOST. While the revised scheme demonstrates consistent results with measurements for surface fluxes, some concerns regarding its physical logic and applicability to complex and heterogeneous surfaces are raised. Therefore, I recommend a major revision before considering this manuscript for publication in Atmospheric Chemistry and Physics.

Response: We kindly thank the anonymous reviewer for your appreciable comments that helped us greatly improve the manuscript's quality. All the comments have been seriously considered, and the manuscript is revised accordingly to meet the publication standard in Atmospheric Chemistry and Physics.

**[Major Comments]**

1. There are doubts about whether the application of MOST for revising domain-averaged flux calculations is suitable for numerical simulations on complex and heterogeneous underlying surfaces. The authors should thoroughly discuss the limitations and potential challenges of applying MOST in such scenarios.

Response: Thank you for your valuable comment and concern regarding MOST's application for revising domain-averaged flux calculations over heterogeneous underlying surfaces.

It's essential to recognize that MOST originally focused on isotropic turbulence and homogeneous surfaces. Our study primarily aimed to investigate the potential of using MOST as a constraint to rectify surface heat fluxes in LES models over a heterogeneous oasis surface.

While our results showed promising improvements in aligning fluxes with MOST principles, dealing with complex and heterogeneous surfaces introduces uncertainties and challenges about our new approach. Addressing this matter may entail delving into the inertial sublayer, which could potentially extend beyond the initial grid points near the surface. Should we pursue this avenue in the future, a carefully devised strategy will be necessary to handle grid points within the viscous sublayer, where surface heterogeneity significantly influences outcomes.

These limitations and challenges have been added to the revised manuscript. Please see details in the revised manuscript in Lines 338-342.

2. The calculation of flux in equations 3 and 4 should involve the average of the product of turbulent fluctuations. Please clarify and correct this discrepancy.

Response: Done. Please see details in Lines 83-84 in the revised manuscript.

3. Please clarify whether the term in Equations 5 and 6 should be denoted as Hles or Hles,sg.

Response: $H_{les}$. Corrected in the revised manuscript. Please see details in Lines 87-89 in the revised manuscript.

4. The manuscript lacks adequate information on the field observation experiments used for simulation validation.

Response: Information on the field observation experiments have been added to the revised manuscript. We employed the multi-scale evapotranspiration flux observation datasets over heterogeneous land surfaces in the Heihe River Basin from HiWATER (Liu et al., 2011; Li et al., 2013). These datasets encompassed observations from various sites, including the Daman Site (38.85°N, 100.37°E, 1556.00 m ASL), Village Site (38.85°N, 100.35°E, 1561.87m ASL), Orchard Site (38.84°N, 100.36°E, 1559.63m ASL), and radiosonde sounding observations from the Zhangye National Climate Observatory (39.08°N, 100.27°E, 1556.06 m ASL) (Fig. 3). The Daman, Village, and Orchard Sites were situated within the agricultural fields of the Daman Irrigation Area in Zhangye City, China, featuring maize fields, villages, and orchards as representative land surfaces, respectively. Several sensors recorded meteorological data at varying heights above the ground. Specifically, wind speed and wind direction sensors were installed at heights of 3, 5, 10, 15, 20, 30, and 40 m, all oriented northward. An air pressure sensor was situated 2 m above the ground. Additionally, a four-component radiometer was mounted at 12 m, facing south. Soil temperature probes were deployed at the soil surface (0 cm) and depths of 2, 4, 10, 20, 40, 80, 120, and 160 cm, all located 2 m south of the meteorological tower and oriented southward. Soil moisture sensors were buried at depths of 2, 4, 10, 20, 40, 80, 120, and 160 cm, all positioned 2 m south of the meteorological tower. Please see details in Lines 155-166 in the revised manuscript.

Important details, such as the height and orientation of the eddy covariance (EC) system, preprocessing procedures for turbulence data, and the average time scale used for obtaining turbulent fluctuations, are missing. These omissions may significantly affect the observed flux values' magnitude and the comparison with simulated results. Additionally, the height of the measured turbulence data used for comparison with simulated results should be specified, as it can impact the comparison effect, particularly in heterogeneous cases.

Response: The eddy covariance (EC) system at the Daman site, Village site, and Orchard site were mounted at 4.5 m, 6.2 m, and 7.0 m, respectively, with all systems operating at a sampling frequency of 10 Hz. These systems were consistently oriented northward, and the distance between the sonic anemometer (CSAT3) and the $CO_2/H_2O$ analyzer (Li7500A) was maintained at 17 cm, 20 cm, and 0 cm, respectively. The collected EC data underwent rigorous preprocessing to ensure data quality. The EC data were initially temporally aggregated into 30-minute intervals to facilitate subsequent analysis. Furthermore, in line with the methodology outlined by Foken and Wichura (1996), the quality of observational data underwent a classification process that stratified data quality into distinct levels based on criteria such as Δst (stationarity) and integral turbulent characteristics (ITC). Only data within class 1 represented high-quality data (level 0: Δst < 30 and ITC < 30) were considered. A five-step data quality control process was also implemented for the half-hourly flux data. These steps involved: eliminating data obtained during periods of sensor malfunction, characterized by anomalous diagnostic signals and automatic gain control values exceeding 65; rejecting data collected within 1 hour of precipitation events; discarding incomplete 30-minute data segments if missing data accounted for more than 3% of the raw 30-minute record; excluding data acquired during nighttime hours when the friction velocity ($u_*$) fell below 0.1 m s$^{-1}$ (Blanken et al., 1998); considering only wind directions ranging from 315° to 0° and 0° to 45° to mitigate potential influences from adjacent EC sensors or environmental factors such as nearby brackets. Please see details in Lines 167-181 in the revised manuscript.

The observed sensible and latent heat flux, averaged from three EC sites, was employed as the benchmark for evaluating sensible and latent heat flux by LES. The simulated sensible and latent heat flux at a height of 10 m was juxtaposed with these observations for validation purposes. Please see details in Lines 214-216 in the revised manuscript.

References:

Blanken, P., Black, T., Neumann, H., Hartog, C., Yang, P., Nesic, Z., Staebler, R., Chen,W., Novak, M. : Turbulence flux measurements above and below the overstory of a boreal aspen forest. Boundary-Layer Meteorol, 89, 109-140, https://doi.org/10.1023/A:1001557022310, 1998.

Foken, T., Wichura, B.: Tools for quality assessment of surface based flux measurements. Agric. For. Meteorol, 78, 83-105, https://doi.org/10.1016/0168-1923(95)02248-1, 1996.

Liu, S. M., Xu, Z. W., Wang, W. Z., Jia, Z. Z., Zhu, M. J., Bai, J., and Wang, J. M.: A comparison of eddy-covariance and large aperture scintillometer measurements with respect to the energy balance closure problem, Hydrol. Earth Syst. Sci., 15, 1291–1306, https://doi.org/10.5194/hess-15-1291-2011, 2011.

Liu, S., Xu, Z., Zhu, Z., Jia, Z., Zhu, M.: Measurements of evapotranspiration from eddy-covariance systems and large aperture scintillometers in the Hai River Basin, China. Journal of Hydrology, 487, 24-33, doi: http://dx.doi.org/10.1016/j.jhydrol.2013.02.025, 2013.

Whether it is a homogeneous case or a heterogeneous case, α is close to 1, and the difference in α between the two experiments selected by the authors is not significant.

Response: We appreciate your feedback and apologize for any lack of clarity in our previous description. $\alpha_j$ represents the efficiency factor of $H_{\text{most}}$ for land use type $j$. To improve clarity, we have revised the description in Section 2.2 of our manuscript. Please refer to the revised manuscript in Lines 110-145 for a more detailed explanation.

For further clarification on the methodology used to derive these $\alpha$ values, we have provided comprehensive details in the Supplementary Material section of our manuscript.

Commonly speaking, α seems to be related to roughness.

Response: Thank you for your suggestion. You seem to make a valid point. The determination of $\alpha$ is not the primary focus of this study, but it may be related to roughness. We will consider addressing this aspect in future research.

5.The manuscript requires a more comprehensive discussion and introduction of the physical significance of the proposed revised scheme. Improving of the simulation effect seems to be an inevitable result, as the author forces the correction of regional average flux towards the MOST results. It seems more like a mathematical correction. More physical insights are needed to strengthen the manuscript's arguments beyond a purely mathematical correction.

Response: More physical insights strengthen the manuscript's arguments beyond a purely mathematical correction. Please see the details as follows:

The estimation of surface fluxes for LES models is usually based on the MOST. the near-surface diffusivity and viscosity estimated by the MOST-based schemes often diverge from those derived from LES subgrid closures, causing inconsistencies between them (Redelsperger et al., 2001).

To deal with this inconsistency problem in LES models, a strategy is proposed by Shao et al. (2013) to estimate surface fluxes based on the subgrid closure scheme. This strategy ensures that the surface flux estimates and subgrid closure are on a consistent physical basis. However, this strategy requires an extrapolation of eddy diffusivity and viscosity to the surface and thus local surface parameters (e.g., local roughness length), and it is not clear whether the surface fluxes estimated this way satisfy the MOST on the scale for which the theory works well.

Figure 1a shows the wavenumber ranges represented by RANS and LES models, while Fig. 1b visually demonstrates that the MOST parameterization, suitable for RANS models, may not hold true for LES models. In the constant flux layer, such as the sensible-heat flux and temperature profiles, the relationship can be well approximated with the MOST for RANS models, but deviates when applied to LES models. Utilizing MOST-based surface-flux parameterizations contradicts the MOST and introduces internal inconsistencies within LES models, manifesting as disparities between MOST-based estimations of eddy viscosity and diffusivity and those derived from LES subgrid closure. However, if we compute surface fluxes locally as outlined in Shao et al. (2013) and integrate the fluxes over a sufficiently large domain for which the MOST works well, then it is required that

$$\frac{1}{N}\sum_1^N H_{\text{les}}^i = H_{\text{most}} = -\rho c_p K_h \frac{\partial \overline{T}}{\partial z}, \qquad (1)$$

where $H_{\text{les}}^i$ is the surface sensible heat flux estimated by LES for grid cell $i$, $N$ is the total number of grid cells in the domain and $\overline{T}$ is the average temperature over the domain. Eq. (1) is not warranted if the fluxes are simply computed as stated in Shao et al. (2013). Thus, a macroscopic constraint needs to be applied to the local surface flux estimates to ensure adherence to Eq. (1).

This study presents a novel approach for surface flux calculation for LES models. This approach comprises two components. First, it calculates LES sub-grid fluxes using eddy viscosity and diffusivity estimates derived from the LES closure scheme

without invoking the MOST while considering local turbulence characteristics. Second, it employs a macroscopic constraint to ensure that fluxes averaged over the LES domain, which corresponds to scales suitable for MOST application, align with the MOST principles. This scheme requires only LES simulated variable, effectively addresses the previously mentioned limitations and ensures that LES flux estimates to be independent of model resolution. It facilitates the knowledge transfer from LES to RANS models.

As far as flux estimates in LES models are concerned, the use of eddy viscosity and diffusivity derived from the LES turbulent closure accounts for the heterogeneities on scales larger than the LES model resolution. Hence, the MOST-r scheme already integrated the basic ideas of the two-part eddy viscosity model of Sullivan et al. (1994), without invoking the assumption of reduced turbulent contribution to subgrid eddy viscosity and diffusivity. This enables the MOST-r scheme to better model the surface flux patterns. Instead of attempting to derive a general scale-invariant scheme for flux estimates, we provided a scheme that is both simple and effective for LES models.

These contents have been added to the revised manuscript. Please see details in Lines 31-152 in the revised manuscript.

6. The writing style and clarity of the manuscript should be further improved for better readability and understanding.

Response: We have carefully considered your suggestions and have made the necessary revisions to enhance the writing style and clarity of the manuscript. We have taken steps to improve the readability and overall understanding of the content, making sure that our research is effectively communicated to our readers. The manuscript is revised accordingly to meet the publication standard in ACP.